# Facial Expression Recognition: One Attention-Modulated Contextual Spatial Information Network

**DOI:** 10.3390/e24070882

**Published:** 2022-06-27

**Authors:** Xue Li, Chunhua Zhu, Fei Zhou

**Affiliations:** 1College of Information Science and Engineering, Henan University of Technology, Zhengzhou 450001, China; lixue202082@163.com (X.L.); hellozf1990@163.com (F.Z.); 2Henan Key Laboratory of Grain Photoelectric Detection and Control, Henan University of Technology, Zhengzhou 450001, China; 3Key Laboratory of Grain Information Processing and Control, Henan University of Technology, Ministry of Education, Zhengzhou 450001, China

**Keywords:** facial expression recognition, features extraction, spatial information, neural network, deep learning

## Abstract

Facial expression recognition (FER) in the wild is a challenging task due to some uncontrolled factors such as occlusion, illumination, and pose variation. The current methods perform well in controlled conditions. However, there are still two issues with the in-the-wild FER task: (i) insufficient descriptions of long-range dependency of expression features in the facial information space and (ii) not finely refining subtle inter-classes distinction from multiple expressions in the wild. To overcome the above issues, an end-to-end model for FER, named attention-modulated contextual spatial information network (ACSI-Net), is presented in this paper, with the manner of embedding coordinate attention (CA) modules into a contextual convolutional residual network (CoResNet). Firstly, CoResNet is constituted by arranging contextual convolution (CoConv) blocks of different levels to integrate facial expression features with long-range dependency, which generates a holistic representation of spatial information on facial expression. Then, the CA modules are inserted into different stages of CoResNet, at each of which the subtle information about facial expression acquired from CoConv blocks is first modulated by the corresponding CA module across channels and spatial locations and then flows into the next layer. Finally, to highlight facial regions related to expression, a CA module located at the end of the whole network, which produces attentional masks to multiply by input feature maps, is utilized to focus on salient regions. Different from other models, the ACSI-Net is capable of exploring intrinsic dependencies between features and yielding a discriminative representation for facial expression classification. Extensive experimental results on AffectNet and RAF_DB datasets demonstrate its effectiveness and competitiveness compared to other FER methods.

## 1. Introduction

Facial expression is an essential skill for humans. In daily life, human emotions are mainly conveyed in three ways, namely language, voice, and facial expressions. Some studies have shown that about 55% of emotional information comes from facial expressions, 38% from voice, and 7% from language [1]. As a result, the technology of facial expression recognition (FER) has attracted extensive attention in scenarios that require the assistance of non-verbal communication such as human-computer interaction, mental treatment, and educational evaluation [2,3,4]. Although some existing methods have made substantial progress, they are not well adapted to variable environments. Concretely, in the presence of occlusion and pose variation, some invisible facial areas and the subtle changes of the face make it hard to distinguish different categories of expressions. Therefore, achieving accurate in-the-wild FER is still a challenging task.

Up to now, many studies have been conducted on FER. They can be roughly divided into (i) classical methods based on traditional artificial features and (ii) popular methods based on deep learning (DL). Driven by various hand-crafted descriptors, the classical methods mainly included local binary patterns (LBP) [5], scale-invariant feature transform (SIFT) [6], and histogram of oriented gradients (HOG) [7]. These descriptors are developed to extract geometric or textural features from facial images to train classifiers. At first, LBP is used for feature extraction because of its computational simplicity and illumination invariance [8], which was invested to extract expression features from facial images divided into several blocks in [9] to deal with the multi-view FER issue. Then, a multi-class support vector machine (SVM) was used for classification. As the LBP-based methods were readily affected by the noise, some variants of LBP were proposed to improve its robustness. The local ternary pattern (LTP) was proposed by Tan [10], and the Scharr operator was added by Tapamo [11], which partially solved the problem in the presence of noise. Their disadvantage is that the selection of threshold was difficult. In [12], Jabid et al. proposed a local directional pattern (LDP) based on the edge responses in eight different directions, but its performance depends on the number of edge response directions [13]. Then, SIFT is invariant for the rotation and scaling of images [14]. In [15], Ryu et al. a facial descriptor called local directional texture pattern (LDTP) was presented for taking advantage of the edge patterns, which encoded the information of emotion-related features. Zheng et al. used SIFT features extracted from landmarks of certain locations of each facial image to describe the facial image for expression recognition in [16,17], such as the landmarks near the mouth and eyes. However, some unnecessary information is also provided by the SIFT descriptor. In addition, the HOG descriptor earned facial features from each pair of mouth and eye regions of the face in [18]. Then a k-nearest neighbor (KNN) classifier was invested in classifying facial expressions. Subsequently, Wang et al. proposed a multi-orientation gradient (MO-HOG) for calculating features [19]. Nevertheless, the approach takes a significant amount of time [20]. Depending on prior knowledge, pre-defined descriptors are not flexible enough to represent facial expression images that vary from different collection environments. Furthermore, the hand-crafted features are sensitive to occlusion, illumination, and pose variation.

In recent years, deep learning (DL) methods, especially convolution neural networks (CNN), have performed great possibilities in FER tasks [21,22]. Compared with classical methods, the DL methods not only integrate the two processes of feature extraction and classification but also actively acquire knowledge from numerous data through neuronal computing. The CNN-based methods for FER, which have powerful feature extraction capabilities [23,24,25], can automatically acquire facial expression information in deep layers. Some FER methods based on CNN and its variants were presented to handle occlusion and pose variation. Li et al. constituted a patch-based attention CNN (pACNN) by combining CNN with an attention mechanism [26], in which each feature map was decomposed into several patches according to the positions of related facial landmarks to perceive the visible regions and reweigh each patch by its importance. However, this method relies on robust face detection and facial landmark localization. In addition, a deep locality preserving CNN (DLP-CNN) was proposed in [27], which preserved the local closeness and maximized the inter-class distinction to enhance the discrimination of deep features. The specific approach is to improve the discrimination ability of deep features by adding a new supervised layer named locality preserving loss (LP Loss) on the fundamental architecture. In 2020, Lian et al. analyzed the contribution from different facial regions, including nose areas, mouth, eyes, nose to mouth, nose to eyes, and mouth to eyes areas, answering the emotion recognition confidence based on partial faces [28]. Moreover, the three sub-networks comprised of CNNs with different depths were ensembled together to constitute the whole model in [29]. The sub-network with more convolutional layers extracted local details such as the features of the eyes and mouth, while the sub-network with less convolutional layers focused on the macrostructure of the input image. However, because the training set of each sub-network is the same, the problem of over-fitting is easy to produce. In [30], a feature selection network (FSN) extracted facial features by embedding a feature selection mechanism inside the CNN, which filtered irrelevant features and emphasized correlated features according to learned feature maps. Despite these methods being successful, they describe potential relation between deep features incompletely. In addition, the subtle discrimination of different expressions is not finely captured by them.

From our observations, it will be more effective if a holistic representation of facial expression information is learned, with the long-range dependency of expression features in the facial information space sufficiently described, which is beneficial for the recognition of multiple facial expressions with occlusion or pose variation in the wild. Alternatively, to enhance inter-class differences in expressions, long-range dependent features extracted from the network should be modulated across the spatial-channel dimension for subtle information refinement, accurately discriminating different categories of expressions. In this paper, a novel model termed attention-modulated contextual spatial information network (ACSI-Net) is proposed for in-the-wild FER. A contextual convolutional residual network (CoResNet) and coordinate attention (CA) modules are combined in our method. Primarily, a contextual convolutional residual network (CoResNet) is constructed by employing contextual convolutions of different levels, which integrates spatial information of the face to obtain a holistic representation of facial expression information. Next, the coordinate attention (CA) modules are embedded into different stages of CoResNet, which are utilized to weight features with long-range dependency across channels and spatial locations, focusing on detailed information on facial expression. In each stage, the information on facial expression acquired from contextual convolution blocks is first modulated by the corresponding CA module, and then flows into the next layer. Lastly, to highlight salient facial regions related to expression, a CA module is following the whole network which produces attentional masks to multiply by input features. The contributions of our work can be summarized as follows:Aiming at the in-the-wild FER task, we propose ACSI-Net. Different from some existing methods, ACSI-Net is able to focus on salient facial regions and automatically explore the intrinsic dependencies between subtle features of expression.To generate a holistic representation, a CoResNet is constructed for long-range dependent expression features extraction by supplying contextual convolution (CoConv) blocks in the main stages of the residual network (ResNet) to integrate spatial information of the face.The CA modules are adopted to adaptively modulate features, pushing the model to retain relevant information and weaken irrelevant ones. Extensive experiments conducted on two popular wild FER datasets (AffectNet and RAF_DB) demonstrate the effectiveness and competitiveness of our proposed model.

## 2. Related Work

In the past few years, a lot of efforts have been made on FER under different conditions. Since our method benefits from deep learning (DL), we briefly review some current DL-based methods for FER that are closely related to our research. Two aspects are elaborated on in the following.

On the one hand, to extract subtle features of facial expression in deep networks, some methods improved the performance of FER through complementary layers or structures of branches. For example, Zhao et al., designed a symmetric structure to learn multi-scale representation in residual blocks and keep facial expression information at the element level [31]. The slide-patch (SP) was proposed in [32] to slide self-calculated windows on each feature map to extract global features of facial expressions. Fan et al. [33] modeled a hierarchical scale network (HSNet), in which the scale information of facial expression images was enhanced by a dilation convolution block. In [34], a dual-branch network was projected with one branch using CNN to capture local marginal information and the other applying a visual transformer to obtain compact global representation. Wang et al. constructed an architecture similar to U-Net as an attention branch to highlight subtle local facial expression information [35]. The local representation was obtained by a multi-scale contractive convolutional network (CCNET) in [36]. A multi-layer network after CNN architecture was exploited in [37] to learn higher-level features for FER.

On the other hand, some attention modules were utilized in parts of works to further enhance the representational ability of features. For instance, Xie et al. highlighted the salient features related to expressions by a salient expressional region descriptor (SERD) [38]. A spatial-channel attention network (SCAN) was designed in [39] to obtain local and global attention at the different spatial positions from different channels, jointly optimizing features. In [40], Sun et al. considered attention at the pixel level to learn a weight for each pixel of channels. A discriminative attention-based convolution neural network (DA-CNN) was proposed in [41] to generate comprehensive representations, which focus on salient regions by spatial attention. Additionally, refs. [42,43] used spatial attention to capture the face area of an image for FER.

Moreover, it is highlighted in [44,45,46] that the importance of contextual information in the visual systems. Specifically, the extraction of the semantic meaning of each local region of an image is only possible if information from other regions is considered. Hence, the contextual information should be integrated for feature refinement. What is more, CA mechanism considers a more efficient way of capturing positional information and channel-wise relationships to augment the feature representations [47]. Motivated by these advanced studies, in our work, the contextual information cooperating with the CA mechanism is employed to perform in-the-wild FER effectively.

## 3. The Proposed Method

The overall construction of the proposed ACSI-Net is shown in Figure 1, which contains a CoResNet for extracting features with long-range dependency and four attention modules for refining subtle features and highlighting salient expressional regions. In CoResNet, there are four main stages, each of which consists of contextual convolution blocks with different levels. These levels can be adjusted according to the size of the input feature maps. In addition, the four attention modules are integrated into CoResNet to modulate the facial information acquired at each stage. The attention module at the end of the network generates an attention mask, which is used for element-wise multiplication with features extracted by CoResNet to obtain salient features.

### 3.1. CoResNet for Feature Extraction

Long-range dependency is crucial for FER. The features with long-range dependency capture local facial details and describe global facial semantic information in the deep network for FER. In our method, a contextual convolution block was introduced in each residual block to obtain long-range dependent features of facial expression in the global scope.

As shown in Figure 2, the feature maps $M^{in}$ were received by a contextual convolution (CoConv) block which applies different levels $L=\left\{1,2,3,\ldots ,n\right\}$ with different dilation ratios $D=\left\{d_{1},d_{2},d_{3},\ldots ,d_{n}\right\}$, that is, the CoConv block of level=i have dilation ratios $d_{i},\forall i\in L$. From i=1 to n, the dilation ratio increases gradually, which can broader contextual information increasingly. The facial information of local details was captured by contextual convolution kernels with lower dilation ratios, and the contextual information of expression in global space was charged by kernels with higher dilation ratios incorporating. At level=i, the CoConv block provides multiple feature maps $M^{out_{i}}$, for all i∈L, each feature map has the width of $W^{out}$ and the height of $H^{out}$.

A CoConv block contains convolution kernels with dilation ratios of different levels which extract long-range dependent features of facial expression images through receptive fields of different sizes. The convolution kernel size is usually constant in a basic CNN that employs fixed dilation ratios because the increase in size will increase the number of parameters and calculation time [48]. All convolutions in CoConv blocks are independent and allow parallel execution like the standard convolution layer. In contrast, the CoConv block can integrate contextual information while maintaining a similar number of parameters and computational costs. Therefore, the CoConv block should be exploited as a proper substitute for the standard convolution layer. The learnable parameters (weights) and the floating-point operations (FLOPs) for contextual convolution are the same as those of standard convolution and can be calculated as:(1)$$params=M^{in}\cdot K^{w}\cdot K^{h}\cdot M^{out}$$
(2)$$FLOPs=M^{in}\cdot K^{h}\cdot K^{w}\cdot M^{out}\cdot W^{out}\cdot H^{out}$$
where $M^{in}$ and $M^{out}$ represent the number of input and output feature maps, $K^{w}$ and $K^{h}$ represent the width and height of the convolution kernel, and $W^{out}$ and $W^{out}$ represent the width and height of the output feature maps.

Other than some previous works of cascaded networks, in our method, CoConv was directly integrated into residual blocks to construct CoResNet. Referring to previous studies [21,22,23,24,25,26,27,28,29,30], we found that the geometric features of facial images come from the shallower stages and the deeper stages extract semantic features about expressions. Therefore, there are four main stages in CoResNet, each of which has a contextual convolution residual block of the corresponding level, with the lower dilation ratios capturing local detail and the higher dilation ratios describing global information of expression in space. As the farther the layers are from the input, the smaller the size of the feature map will be. The level of the CoConv block in each stage was adapted concerning the size of the feature maps. We set level=4 with different dilation ratios in the first main stage. Then, the second stage uses level=3 in its CoConv layer, and level=2 in the third stage. As the size of the feature map is 7×7 in the final stage, just one standard convolution was used, denoted as level=1. Parameters of different stages are shown in Table 1.

### 3.2. CA Modules for Feature Refinement

To further refine features with long-range dependency, CA modules were embedded in contextual residual blocks of CoResNet, which guided the network to pay attention to significant features. Meanwhile, a CA module following CoResNet was utilized to highlight the salient facial regions. These CA modules retain the expression-related information and weaken irrelevant ones. The structure of the coordinated attention module is shown in Figure 3.

Given the feature map of input, firstly, each channel was encoded through 1D horizontal global pooling and 1D vertical global pooling, respectively. The encoded output of the c-th channel can be formulated as follows:(3)$$y_{c}^{h}(h)=\frac{1}{W}\sum_{0\leq i\leq W}x_{c}(h,i)$$
(4)$$y_{c}^{w}(w)=\frac{1}{H}\sum_{0\leq i\leq H}x_{c}(j,w)$$
where $x_{c}$ denotes the input of the *c-th* channel that waits to be encoded. $y_{c}^{h}$ denotes the encoded output of the c-th channel with a height of h, and $y_{c}^{w}$ denotes the *c-th* channel with a width of w.

The feature maps were aggregated, and then a pair of direction-aware attention maps were returned by (3) and (4). The locations of the salient facial regions were retained to help the network focus on expression-related features more accurately. Next, the two feature maps were connected and fed into a 1×1 convolution function F, which can be formulated as:(5)$$f=\delta \left(F\left(\left[y^{h},y^{w}\right]\right)\right)$$
where   refers to concat operation across the spatial dimension, δ is the sigmoid nonlinear activation function, and f∈RCr×(H+W) is the feature maps encoding spatial information in the horizontal and vertical directions. An appropriate reduction rate was adopted to cut down the number of channels for lowering the complexity of the model. Sequentially, f is disentangled into two separate tensors fh∈RCr×H and fw∈RCr×W along the two spatial dimensions, and then, the two 1×1 convolutions $F_{h}$ and $F_{w}$ are employed into $f^{h}$ and $f^{w}$, which have the same number of channels, can be expressed as:(6)$$m^{h}=\delta \left(F_{h}\left(f^{h}\right)\right)$$
(7)$$m^{w}=\delta \left(F_{w}\left(f^{w}\right)\right)$$
where $m^{h}$ and $m^{w}$ are the weights of the attentive mask. Finally, the output of the coordinate attention module was calculated as follows:(8)$$z_{c}(i,j)=x_{c}(i,j)\times m_{c}^{h}(i)\times m_{c}^{w}(j)$$

## 4. Results and Discussion

### 4.1. Datasets

In contrast with lab-controlled datasets such as JAFFE, CK+, and MMI, wild FER datasets are collected in uncontrolled conditions offering diversity across pose, occlusion, and illumination. AffectNet and DAF_DB are two widely used wild datasets in FER research. Details of experimental datasets are exhibited in Table 2.

**AffectNet** [49] is the largest dataset with 1M facial images acquired from the Internet, about 420 K of which are manually annotated. It contains AffectNet-7 and AffectNet-8 (adding “contempt” category). In the experiments, we use AffectNet-7, including six basic facial expressions and neutral. There are about 280 K images for training and 3500 images for testing.

**DAF_DB** [27,50] contains about 30 K real-world facial images. Based on the crowd-sourcing techniques, each image has been independently labeled by about 40 trained annotators. There is a single-label subset with seven categories of basic emotions and a two-label subset with twelve categories of compound emotions. For our experiments, the single-label subset is used.

### 4.2. Implementation Details

All input images are aligned and resized to 256×256, the standard stochastic gradient descent (SGD) optimizer is adapted with a momentum of 0.9 and a weight decay of $5\times 10^{-4}$ during training; the training data are augmented by extracting five random crops of size 224×224 (one central and four from corners, as well as their horizontal flips). At the testing time, the central crops are fed into a trained model on AffectNet-7 and DAF_DB for 60 epochs; the initial learning rate is 0.01 with a reduction in a factor of 10 every 20 epochs; the batch size is set to 32 for AffectNet-7 and 16 for DAF_DB, and in experiments, the model is trained with Pytorch on NVIDIA GeForce GTX 1650 GPU with 16 GB RAM. Notably, the pre-training strategy is adapted for saving the total training time and can obtain superior performance [51]. In this paper, the proposed model will be pre-trained on a face dataset MS-CELEB-1M [52], and then fine-tuned on FER datasets, owing to the similarity between the domain of FER and the face recognition (FR) task.

### 4.3. Ablation Studies

In this section, the ablation studies on CoResNet and CA modules are conducted, respectively, to verify the effectiveness of our ACSI-Net. At first, the CoResNet compared with ResNet is investigated. Then, we evaluate the performance of the CA module assigned at different locations in the network.

#### 4.3.1. Ablation Study of CoResNet

The performance of CoResNet is displayed in Table 3, in which the number of model parameters, FLOPs, test time of one image, and recognition accuracy are compared with ResNet. From Table 3, we can observe that the proposed CoResNet can provide the higher recognition accuracy than ResNet on both RAF-DB and AffectNet-7 datasets with the same cost. it is owning to that the CoResNet can utilize CoConv to extract long-range dependent features of facial expression in contextual space, and the learnable parameters (weights) and floating-point operations (FLOPs) of the CoConv are equal to standard convolution. In addition, to ensure the lightweight of the network, the network layers are designed as 18 in our experiments.

To better explain the effect of CoConv, we visualize the feature distribution of samples from RAF-DB and AffectNet-7 datasets under ResNet and CoResNet, as shown in Figure 4. The visualizations are implemented by using t-SNE [53], which is a widely used tool for visualizing high-dimensional data [54]. We can see that features extracted from ResNet (Column 1) are not easily distinguishable for facial expression classes. In contrast, the features extracted from CoResNet (Column 2) tend to shape several clusters in the space. Inter-class separability is heightened essentially since extensive contextual spatial information of face is integrated to refine features by CoConv in CoResNet.

#### 4.3.2. Ablation Study of CA

As described in Section 3.2, the coordinate attention (CA) modules are utilized to weigh the features at two different embedding locations. The two ablation experiments are conducted to evaluate the CA module. In the first experiment, we train CoResNet where the CA module is embedded in each residual block, denoted as CoResNet_CA-a. In the second experiment, CoResNet_CA is trained by embedding the CA module after CoResNet, denoted as CoResNet_CA-b. Table 4 shows the performance of the above models and the proposed ACSI-Net. From Table 4, the ACSI-Net performs better than both CoResNet_CA-a and CoResNet_CA-b on accuracy (%), which means the combination of the four embedded CA modules with different locations is efficient in the proposed ACSI-Net, they can modulate the facial information acquired at each stage. In contrast, there is no significant increase in the runtime and space complexities.

To investigate the performance of our model, we used the class activation map (CAM) [55] to visualize the attention maps generated by our ACSI-Net. Specifically, we resize the attention maps to the same size as the input images and visualize the attention maps to the original image through COLORMAP_JET color [56]. Figure 5 shows the attentional regions of different expression images in RAF_DB. There are seven columns, and each is one of the expression classes. From left to right, the labels of classes are anger, disgust, fear, happiness, sadness, surprise, and neutral. The first row shows the original aligned face image, and the second row shows the result of the ACSI-Net. From these attention maps, we can conclude that our proposed model has the ability to focus on the discriminable regions of the occluded or pose-variable face. There is a phenomenon that the nose and its nearby regions contribute the most to the prediction. That is because the nose and its nearby regions play an important role in discriminating some emotions when the mouth or eyebrows is occluded. In ACSI-Net, the subtle information from these regions is captured and the corresponding salient features are filtered out.

### 4.4. Quantitative Evaluation

On AffectNet-7 and RAF_DB datasets, the recognition accuracy of the proposed ACSI-Net is shown in Table 5, compared with the existing ones including the gACNN [26], the CPG [57], the separate loss [58], the MA-Net [32], the OAENet [35], the DACL [49], and the HSNet [33]. 

From Table 5, on the AffectNet-7 dataset, the proposed ACSI-Net obtains a recognition accuracy of 65.83% which is the state-of-the-art result among the existing ones. Notedly, the recognition accuracy of the ACSI-Net is 86.86% on the RAF_DB dataset, which is inferior to the DACL and outperforms the other ones. In DACL, a sparse deep attentive center loss jointly with softmax loss is adapted to enhance the discriminative power of learned features in the embedding space, and there are more operations of dimension reduction compression, which will increase the network complexity. Comparably, in the propose ACSI-Net, one more efficient way is adapted to embed attention modules for capturing positional information and channel-wise relationships, which can augment the feature representations. In addition, these embedded attention modules can directly modulate feature information with different levels in the contextual space. 

## 5. Conclusions

In this paper, we propose a novel model named ACSI-Net for FER under wild scenarios. The proposed model combines a feature extraction network (CoResNet) and attention modules (CA). In our method, the global features with long-range dependency are extracted by CoResNet and fed into the CA modules at each residual block, which integrates the contextual spatial information of facial expression potentially and highlights expression-sensitive features. Further, a CA module following CoResNet is utilized to adaptively quantify the importance of features to optimize them. Visualizations show that CoResNet is stimulative for the difference of inter-class features, while CA modules can make the network automatically focus on some expression-related areas such as the neighborhood of eyes and mouth. Eventually, ASCI-Net can distinguish diverse expressions. Experimental results demonstrate that the proposed ASCI-Net outperforms the existing comparable methods.

However, the proposed ASCI-Net can still be improved in some aspects. In our future work, we intend to model relationships among facial patches with a vision transformer. By learning relation-aware representations in the global scope, the performance of the network is expected to improve.

## Figures and Tables

**Figure 1 entropy-24-00882-f001:**
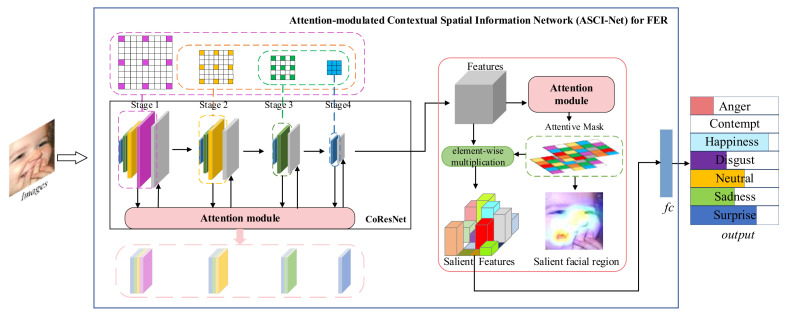
The overall construction of our proposed attention-modulated spatial information network (ASCI-Net).

**Figure 2 entropy-24-00882-f002:**
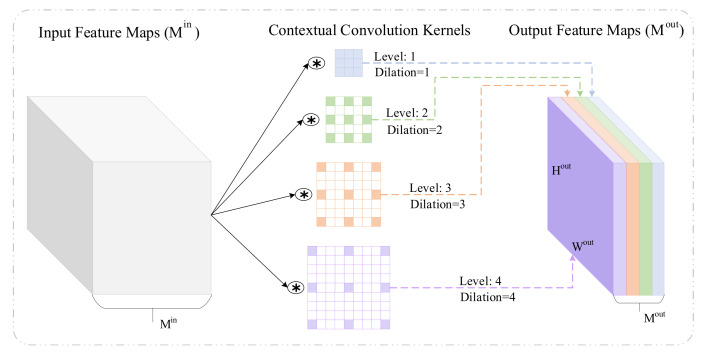
A CoConv block integrating kernels with different dilation ratios in the convolution layer.

**Figure 3 entropy-24-00882-f003:**
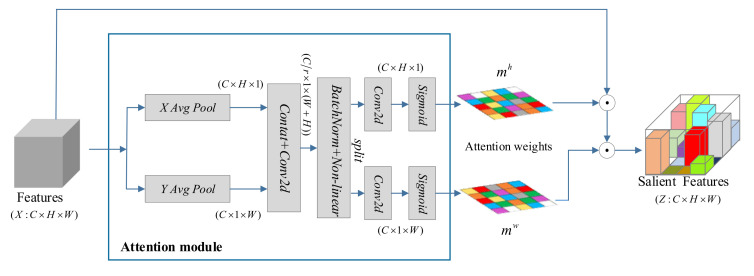
The structure of the coordinated attention module. Here, “X Avg Pool“ and “Y Avg Pool “ refer to 1D horizontal global pooling and 1D vertical global pooling, respectively.

**Figure 4 entropy-24-00882-f004:**
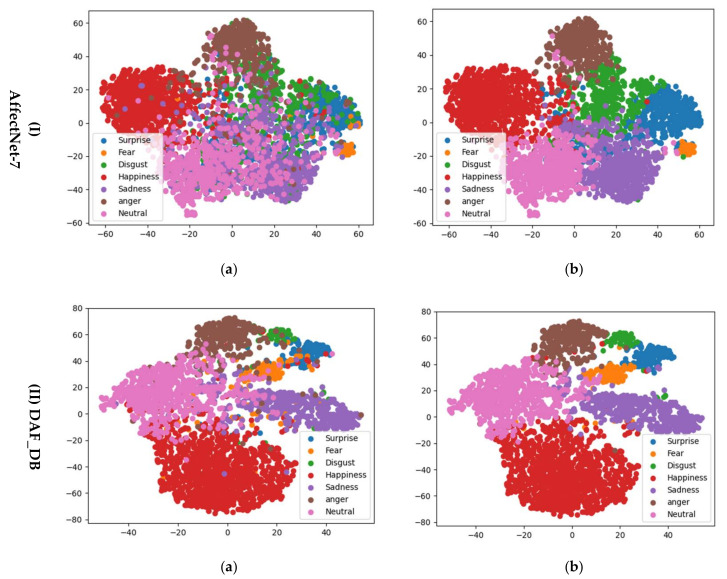
The distribution of deeply learned features under (**a**) “ResNet” and (**b**) “CoResNet” for samples from AffectNet-7 (Row 1) and DAF_DB (Row 2) datasets. As we can see, CoResNet can learn features with more discrimination. Moreover, it is seen that the features extracted from CoResNet tend to shape several clusters in the space.

**Figure 5 entropy-24-00882-f005:**
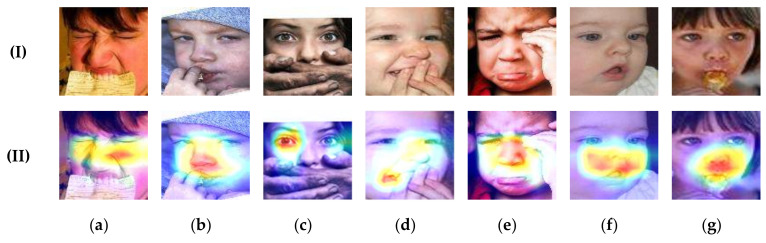
Attention visualization of different facial expressions on some examples from the RAF_DB dataset under the ACSI-Net. (I), (II) denote original facial images and attentive masks to the original image, respectively. (**a**–**g**) denote anger, disgust, fear, happiness, sadness, surprise, and neutral separately.

**Table 1 entropy-24-00882-t001:** Parameters of the CoConv blocks.

Stage	Input Size	Level	CoConv
1	56×56	level=4	$\left[\begin{array}{c} \begin{array}{c} 3\times 3,16,d_{1}=1\\ 3\times 3,16,d_{2}=2 \end{array}\\ \begin{array}{c} 3\times 3,16,d_{3}=3\\ 3\times 3,16,d_{4}=4 \end{array} \end{array}\right]$
2	28×28	level=3	$\left[\begin{array}{c} \begin{array}{c} 3\times 3,64,d_{1}=1\\ 3\times 3,32,d_{2}=2 \end{array}\\ 3\times 3,32,d_{2}=3 \end{array}\right]$
3	14×14	level=2	$\left[\begin{array}{c} 3\times 3,128,d_{1}=1\\ 3\times 3,128,d_{2}=2 \end{array}\right]$
4	7×7	level=1	$\left[3\times 3,512,d_{1}=1\right]$

**Table 2 entropy-24-00882-t002:** Details of experimental datasets, including categories of expressions, number of training and testing samples.

Dataset	Affectnet-7		RAF_DB	
	Train	Test	Train	Test
Anger	24,882	500	705	162
Disgust	3803	500	717	160
Fear	6378	500	281	74
Happy	134,415	500	4772	1185
Sad	25,459	500	1982	478
Surprise	14,090	500	1290	329
Normal	74,874	500	2524	680
Total	283,901	3500	12,271	3068

**Table 3 entropy-24-00882-t003:** The performance of CoResNet and ResNet, including the number of model parameters, FLOPs, test time of an image, and recognition accuracy (%).

Model	Params	GFLOPs	Time/s	Accuracy (%)	
				RAF-DB	AffectNet-7
ResNet	11.69	1.82	1.32	85.88	63.82
CoResNet	11.69	1.82	1.32	86.86	65.83

**Table 4 entropy-24-00882-t004:** The performance of the CA module at different network locations, including the number of model parameters, FLOPs, test time of an image, and recognition accuracy (%).

Model	Params	GFLOPs	Time/s	Accuracy (%)	
				RAF-DB	AffectNet-7
CoResNet	11.69	1.82	1.32	86.29	64.38
CoResNet_CA-a	11.72	1.82	1.33	86.45	65.16
CoResNet_CA-b	11.75	1.82	1.35	86.52	65.60
**ACSI-Net**	11.78	1.82	1.38	**86.86**	**65.83**

**Table 5 entropy-24-00882-t005:** The recognition accuracy (%) of different models on RAF_DB and AffectNet-7.

Method	Year	RAF_DB	AffectNet-7
gACNN [26]	2018	85.07	-
CPG [57]	2019	-	63.57
Separate Loss [58]	2019	86.38	-
MA-Net [32]	2021	86.34	64.54
OAENet [35]	2021	86.50	-
DACL [51]	2021	**87.78**	65.20
HSNet [33]	2022	86.67	-
**ACSI-Net(ours)**		86.86	**65.83**

## Data Availability

Publicly available datasets are analyzed in this study. The data can be found here: http://mohammadmahoor.com/affectnet/ accessed on 22 May 2021; http://www.whdeng.cn/RAF/model1.html accessed on 24 June 2021.

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
