# Peer review of "Facial Expression Recognition: One Attention-Modulated Contextual Spatial Information Network"

_entropy, 2022, doi:10.3390/e24070882_

Round 1

Reviewer 1 Report

The paper proposes a contextual convolutional residual network (CoResNet) constituted by arranging contextual convolution (CoConv) blocks of different levels to integrate facial expression features with long-range dependency, which generates a representation of spatial information on facial expression. The proposed architecture uses AffectNet and RAF_DB datasets to demonstrate its effectiveness and competitiveness compared to other Facial expression recognition (FER) methods. 

Strength:

1. The paper's organization is good, and the proposed methodology exhibits comparable results to the competing methods. 

Weakness and comments:

1. This paper must be checked for grammar, spelling, and many typos. For example, space issues in Line 133. Typos in Line 203 and Fig. 2.

2. Section 3.1, in reference to Fig. 1, must be explained clearly. Please re-write the section explaining each block and module.

3. Line 197, CoConv was defined later in the text.

4. The paper lacks novelty. The motivation of the article is not clear. The results are comparable to existing methods, as shown in Table 5 and Table 6.

5. The algorithm's runtime and space complexities are missing. It needs to be compared with the existing methods in terms of analytical models and simulation results.

Author Response

Dear Reviewer,

Many thanks for your valuable suggestions and time which have improved the quality of our manuscript. Our detailed responses to these comments are in attachment. 

With kind regards,

Xue Li, Chunhua Zhu, Fei Zhou

Reviewer 2 Report

O1) In Figure 2: “Contextual Converlution Kernels”. Do you mean “Convolution”? 

O2) In Equations (3) and (4), what is x_c?

O3) Table 2: There is a clear class imbalance. Were your results affected by it? Did you deal with this problem in any way?

O4) “Furthermore, since the domain of FER is close to the Face Recognition(FR) task, our networks are pre-trained on a face dataset MS-CELEB-1M [52]”: What improvements have you observed with this training approach?

O5) Figure 4 and Table 3: I was expecting a bigger accuracy discrepancy between RestNet and CoResNet judging by the feature distribution shown in Figure 4 (at least in the AffectNet-7 case).

O6) In Subsection 4.3.2, I suggest you placing Table 4 before the second paragraph. 

O7) Figure 5: What did you conclude from these attention maps? In some emotions, it seems that the nose contributes the most for the prediction. Is that expected?

O8) The proposed approach attains a tiny improvement over other methods shown in the paper, but it did not reach state-of-the-art results. In fact, in your own reference “DACL [49]”, you just put their results on AffectNet, however they have also reported an 87.78% accuracy on RAF_DB. Why is it not displayed in Table 6? I’m aware you did not claim to achieve state-of-the-art results anywhere, you just claimed “competitiveness”. However, the way the paper is written may be misleading. Either clearly claim in your paper that you did not achieve state-of-the-art results or cite works that report better results than your proposal (at the very least put all the results reported in your references for both AffectNet and RAF_DB datasets). 

Author Response

(The authors gave the same response as above.)

Round 2

Reviewer 1 Report

The authors addressed all the concerns adequately.